# Dynamic states of population activity in prefrontal cortical networks of freely-moving macaque

Russell Milton [1], Neda Shahidi[1] & Valentin Dragoi [1,2✉]

Neural responses in the cerebral cortex change dramatically between the 'synchronized' state during sleep and 'desynchronized' state during wakefulness. Our understanding of cortical state emerges largely from experiments performed in sensory areas of head-fixed or tethered rodents due to technical limitations of recording from larger freely-moving animals for several hours. Here, we report a system integrating wireless electrophysiology, wireless eye tracking, and real-time video analysis to examine the dynamics of population activity in a high-level, executive area – dorsolateral prefrontal cortex (dlPFC) of unrestrained monkey. This technology allows us to identify cortical substates during quiet and active wakefulness, and transitions in population activity during rest. We further show that narrow-spiking neurons exhibit stronger synchronized fluctuations in population activity than broad-spiking neurons regardless of state. Our results show that cortical state is controlled by behavioral demands and arousal by asymmetrically modulating the slow response fluctuations of local excitatory and inhibitory cell populations.

[1] Department of Neurobiology & Anatomy, McGovern Medical School, University of Texas, Houston, TX 77030, USA. [2] Department of Electrical and Computer Engineering, Rice University, Houston, TX 77005, USA. ✉email: valentin.dragoi@uth.tmc.edu

A prevailing view in neuroscience is that the activity of a neural population is strongly influenced by the state of the animal. Thus, synchronized fluctuations in the responses of simultaneously recorded neurons have been predominantly observed in several sensory cortical areas of rodent brain[1–7]. During sleep and rest, cortical populations are intrinsically synchronized in the low-frequency range[1,8,9], while during wakefulness they are actively desynchronized by cholinergic inputs received from subcortical areas[10–12]. Previous recordings from sensory areas have shown that the degree to which populations are desynchronized depends on behavioral context, which modulates sensory coding and perception[2–4,7,13,14]. In rodent primary sensory areas, low-frequency activity is apparent in inactive, awake animals, and these oscillations are suppressed by active behaviors like whisking and locomotion[2–5,7]. However, despite the prevalence of these ideas, previous investigations of cortical state dynamics at single cell resolution have been performed in rodent sensory cortex while technical limitations have prevented the analysis of dynamic cortical states in larger animals freely moving in their environment for many hours. Importantly, whether behavioral state modulates cortical population activity in higher-order, executive areas is unknown.

## Results

**Integrated wireless system to examine brain states in monkey.** We developed an integrated system for large-scale electrophysiology, eye tracking, and behavioral state classification in freely-moving macaque monkey. We performed wireless recordings using a 96-channel multielectrode array implanted in the dorsolateral prefrontal cortex (dlPFC; area 46[15]) of two freely-moving monkeys (Fig. 1a and Supplementary Fig. 1; sessions lasted 195 min on average). The large 2D geometry (4 × 4 mm) of the multielectrode array allowed us to investigate changes in brain state across a wide range of cortical distances. A wireless, battery-powered headstage transmitted neural signals to a centralized digital signal processor (DSP) and recording computer via an array of eight antennae placed around the experimental cage (Fig. 1a). A wireless eye tracker was used to monitor pupil diameter and eye movements. Animals were fitted with a wireless eye tracker, which uses a transparent lens to reflect the eye image to an on-board camera (Fig. 1b) mounted to the headpost and transmitting oculomotor information to an eye-tracking computer. Animals were initially fitted with a dummy device until they learned to ignore the lens in their field of view and lost interest in touching and potentially damaging the head-mounted eye tracker. Once the animals could consistently wear the dummy eye tracker without touching it, the dummy was swapped with the real device which the monkeys wore without damaging. To encourage active exploration and locomotion, animals had access to buttons on either side of the cage which intermittently dispensed a small reward pellet if pressed. In addition, sessions included a "lights-out" period to encourage animals to sleep. An overhead video of the animal was used to classify behavioral states as active wakefulness, quiet wakefulness, or rest. Periods when the animal was awake, but the lights were off were not included in subsequent analyses. Classification of behavioral states based on the recorded video was done in two stages. First, any intervals during which the animal was in a sleeping posture for >5 min were defined as "rest". All periods during which the lights were on, and the animal was not in a rest posture were further classified into "active" and "quiet" states by processing the overhead video to quantify movement. The videos were processed using a pixel-wise subtraction algorithm that quantifies the frame-to-frame changes in the camera's field of view indicating animal motion. For each session, the average motion during wake state

was used as a threshold to classify each 10 s epoch as active or quiet wakefulness (Fig. 1c). Overall, we analyzed 11 sessions in two 9-year-old adult male animals, which is equivalent to 2027 epochs of active wakefulness, 6182 epochs of quiet wakefulness, and 2943 epochs of rest.

In order to examine how cortical state is impacted by ongoing behavior, two methods for quantifying cortical oscillatory synchrony were used. Initially, we computed the power ratio of local field potentials (LFPs) in the low-frequency (0.5–10 Hz) and high-frequency (20–59 Hz) bands. In rodent models, cortical activation reduces low-frequency LFP power and increases high-frequency LFP power[4,16,17]. Thus, the power ratio between the low (<10 Hz) and high (>20 Hz) LFP bands is commonly computed to quantify cortical state from LFP recordings[17]. At single-neuron resolution, the "synchronized state" is characterized by neuronal membrane potential fluctuations within the low-frequency band, with action potentials being generated on the crests of these low-frequency oscillations[1]. Therefore, we developed a novel method to quantify the power of low-frequency oscillations in spike rasters recorded from a large population of well-isolated single units, referred here as population synchrony index (PSI). Briefly, population firing rate was computed in each 10 s epoch by taking the total number of spikes recorded across the whole population within each 10 ms bin, then converting to spikes per unit by dividing by the number of units and converting to spikes/second by multiplying with a scaling factor of 100. Due to the large number of units, population firing rate is a smooth continuous signal that is suitable for spectral decomposition. PSI is defined as the average of Fourier coefficients of the population firing rate for the 0.5–10 Hz frequency band divided by its mean (in 10 s epochs) (Fig. 1d). Statistical significance of results was determined using nonparametric tests that do not assume an underlying normal distribution, i.e., Wilcoxon signed-rank and Wilcoxon rank-sum for comparison of two groups of unmatched and matched observations. When significant differences among three groups (active, quiet, and rest) were assessed, Kruskal–Wallis and Friedman tests were used to detect differences in unmatched and matched data, respectively (see "Methods").

**Behavioral state and population activity.** Rest significantly increased the LFP power ratio (power in 0.5–10 Hz divided by that in 20–59 Hz), hence reflecting a shift in cortical processing to a more synchronized state[16,18] (Fig. 2a, b; $p < 0.05$, Wilcoxon signed-rank). In individual neurons, synchronous bouts of high and low activity were visible in spike rasters during rest, but not during wakefulness. These ON/OFF dynamics produced a predominantly slow oscillation in the average firing rate of the population (Fig. 1d, e). The magnitude of slow oscillatory activity, quantified by the PSI, was strongly increased in rest relative to wakefulness (Fig. 2c; $p < 0.005$, Wilcoxon signed-rank), and synchronous activity was associated with a significant decrease in the average firing rate of the neural population (Fig. 2d; $p < 0.005$, Wilcoxon signed-rank). These analyses demonstrate that during rest, low-frequency oscillations, previously observed in EEG studies, can be readily detected in the spike rasters of large populations of well-isolated single units. This opens up the intriguing opportunity of exploring these phenomena at a much finer resolution than has been previously possible.

The relationship between cortical desynchronization and wakefulness was further dissected by extracting active and quiet behavioral states. We thus developed an automated system based on the frame-by-frame analysis of the video data to classify each 10 s epoch of wakefulness into active (e.g., locomotion) or quiet state (Fig. 1c, see "Methods"). Relative to rest, active wakefulness

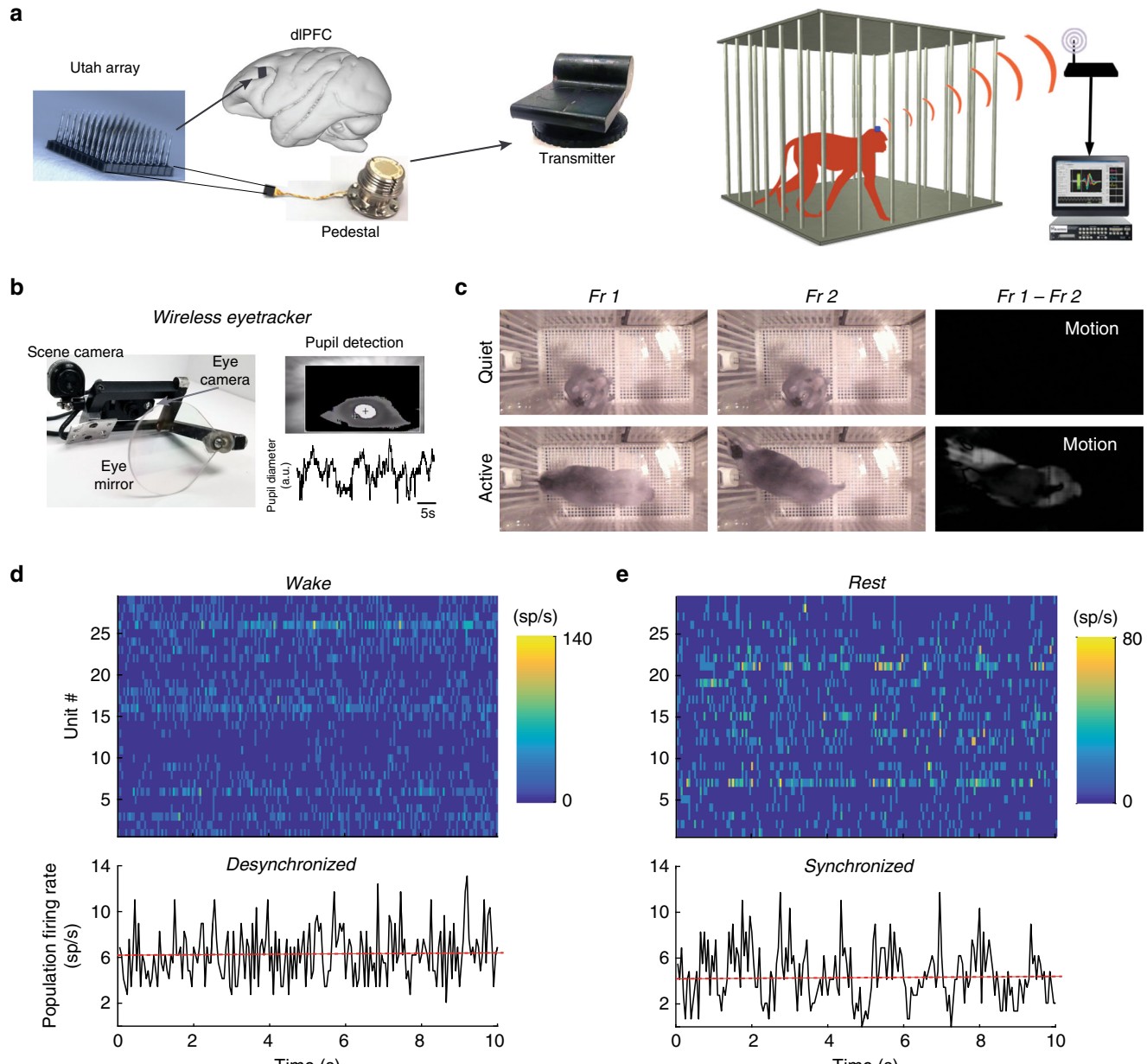

**Fig. 1 Integrated system for monitoring behavioral states and cortical activity. a** Schematic of wireless recording system from a 96-channel microelectrode array in dlPFC of freely behaving macaque. Array image photo credit: Utah Array—© 2020 Blackrock Microsystems, LLC. Monkey cartoon diagram image credit: Reproduced with permission[35]. © IOP Publishing. All rights reserved. Three-dimensional brain cartoon generated with Scaleable Brain Atlas[36-38]. **b** (Left) Schematic of wireless eye tracking system. (Right, top) Example of wirelessly recorded eye image with pupil detection. (Right, bottom) Example of wirelessly recorded pupil diameter. **c** Monkey movement is quantified based on consecutive video frames as the number of pixels that changed in intensity. The average movement during wakefulness for each recording session was used as a threshold to classify 10 s epochs as active or quiet wakefulness. (Top left and top middle) Example of two subsequent frames during quiet wakefulness, (top right) pixel-wise subtraction reveals no motion during this period. (Bottom left and bottom middle) Example of two subsequent frames during active wakefulness. (Bottom right) Pixel-wise subtraction reveals that the monkey is actively behaving during this period. **d** (top) Example raster showing the firing rates of 29 simultaneously recorded single units while the monkey is awake. The spike count in each 10 ms bin was converted to sp/s by scaling by a factor of 100. (Bottom) Population firing rate was computed as the average firing rate for the population within each 10 ms bin (black trace). The red line shows the average population firing rate for the whole 10 s period shown (6.3 sp/s). Population synchrony index (PSI) is 0.0433. PSI quantifies the 0.5–10 Hz oscillations of the population firing rate during the 10 s period. **e** Same as **b**, but recorded while the monkey is resting. Low-frequency fluctuations are apparent in both spike rasters and population firing rate. Average population firing rate is 4.5 sp/s and PSI is 0.0632.

was associated with increased firing rates and lower PSI, indicating a desynchronized state (Fig. 2). Quiet wakefulness represented an intermediate state, with firing rates and PSI values falling between those of rest and active wakefulness. Population firing rates and PSI were significantly different across all behavioral states (Fig. 3a, b; Friedman test, $p < 0.001$; Wilcoxon signed-rank test with Bonferroni correction, $p < 0.001$ for all pairwise comparisons). These results support the idea that the degree of synchrony serves as a readout of cortical state, and may represent the neural substrate of vigilance.

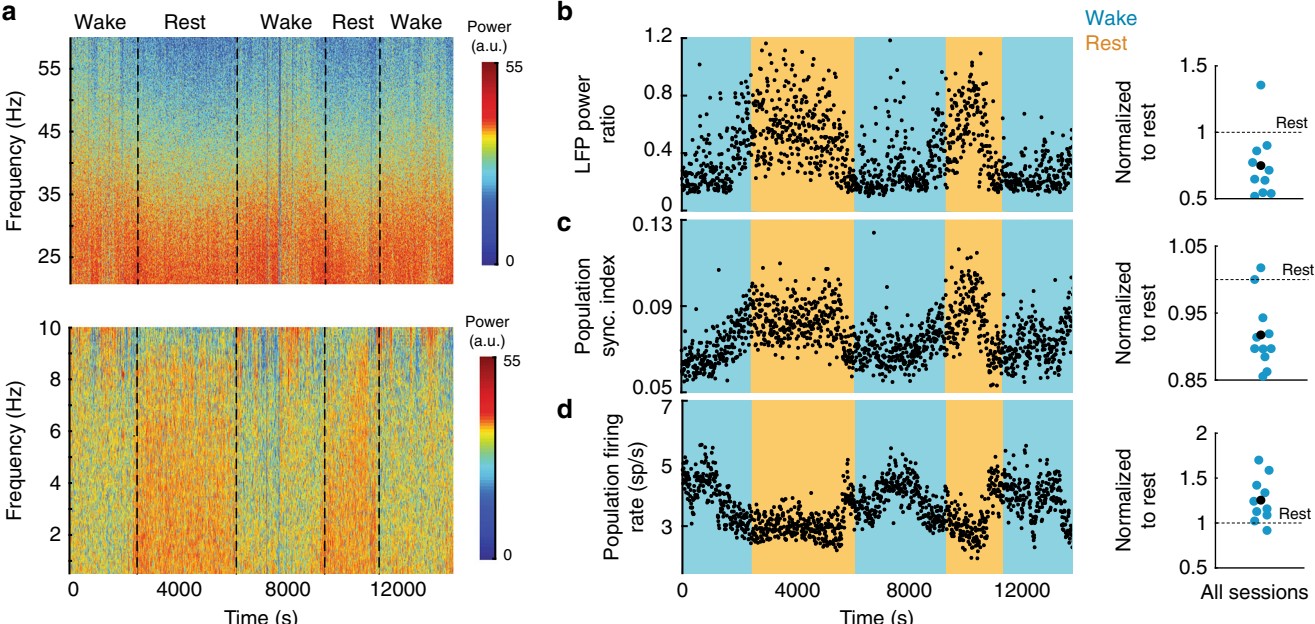

**Fig. 2 Natural rest produces widespread changes in cortical dynamics. a** Spectral analysis of LFPs recorded on the microelectrode array. (Top) High-frequency (20–60 Hz) LFP spectrogram, showing a reduction in power of high-frequency oscillations during rest. (Bottom) Low-frequency (0.5–10 Hz) LFP spectrogram, showing an increase in power of low-frequency oscillations during rest. **b** (Left) LFP synchrony is quantified as the ratio of low to high-frequency LFP power. LFP power ratio is increased during rest. (Right) Summary for all sessions showing LFP power ratio during wake normalized to rest. LFP power ratio during wake is significantly lower than during rest ($p = 0.027$, two-sided Wilcoxon signed-rank, $n = 11$ sessions). On average, LFP power ratio during wake is 75% from that of rest. **c** (Left) PSI is increased during rest. (Right) Summary for all sessions showing PSI during wake normalized to PSI during rest. PSI during wake is significantly lower than during rest ($p = 0.0049$, two-sided Wilcoxon signed-rank, $n = 11$ sessions). On average, the PSI during wake is 91% that of rest. **d** (Left) Population firing rate is reduced during rest. (Right) Summary of all sessions showing average population firing rate during wakefulness normalized to rest. Population firing rate during wake is significantly higher than during rest ($p = 0.0029$, two-sided Wilcoxon signed-rank, $n = 11$ sessions). On average, the population firing rate during wakefulness is 26% higher than during rest.

Our measure of population synchrony index (PSI) has not been previously used to quantify cortical state in freely-moving animals. Therefore, we asked whether the population spiking activity provides a more precise prediction of the behavioral state of the animal than the LFP power ratio. We thus trained a linear support-vector machine to decode the behavioral state of each 10 s epoch based on cortical state as quantified using either PSI or LFP power ratio. PSI and LFP power ratio were z-scored within session to facilitate combining across sessions, and training and test data were randomly sampled such both sets had equal numbers of active, quiet, and rest epochs (see "Methods"). We found that, while both LFP power ratio and PSI can predict behavioral state better than chance level, the PSI-trained decoder was significantly more accurate (Wilcoxon rank-sum test, $p < 0.001$; Fig. 3i). This indicates that our PSI measure provides a more accurate quantification of cortical state than the LFP power ratio.

In sensory cortex, locomotion has been found to increase pupil diameter and reduce low-frequency synchronous oscillations while impacting neural responses in a modality-specific manner[2–4,7]. We examined the key measurable variables associated with behavioral state—motion index, firing rates, PSI, and pupil diameter—to better understand the relationship between physiological and behavioral variables in unconstrained animals. This allowed us to investigate the relationship between cortical state, oculomotor activity, and behavior during different states of wakefulness. Epochs of wakefulness were classified as active or quiet by thresholding the motion index calculated based on the frame-by-frame analysis of animal behavior. We additionally determined how the degree of behavioral activity (i.e., the raw

motion index value) during wakefulness relates to population firing rate, PSI, and pupil diameter. These variables were z-scored for all epochs within each session, and then combined across sessions. We revealed a strong and highly significant correlation between locomotion and spiking activity in dlPFC (Fig. 3c; Pearson's correlation, $R = 0.44363$, $p < 0.0001$). When individual recording sessions were considered, there was a positive correlation between population firing rate and motion index in every session ($p < 0.0001$; Supplementary Fig. 2). We further found a highly significant negative correlation between the z-scored motion index and PSI when combining across sessions (Fig. 3d; Pearson's correlation, $R = 0.11584$, $p < 0.0001$; 8/11 sessions were statistically significant, see also Supplementary Fig. 2). Furthermore, we examined the time course of transitions between active and quiet wakefulness and found that the population firing rate trace closely tracks with motion index (Supplementary Fig. 3). Altogether, these results indicate that population activity in dlPFC strongly depends on ongoing behavioral state during wakefulness.

Autocorrelation analysis reveals that population synchrony, PSI, is highly persistent as it changes slowly in time (over minutes, Supplementary Fig. 4a). This is consistent with the influence of general arousal as the main factor contributing to these changes. We took advantage of the 2D geometry of the multielectrode array to compute PSI within a range of cortical distances, and found greater PSI values at nearby distances while preserving differences in PSI across behavioral states (Supplementary Fig. 4b). A correlate of general arousal is pupil size. Indeed, active behavior generates a state of heightened arousal associated with acetylcholine and norepinephrine release that increases ongoing activity in

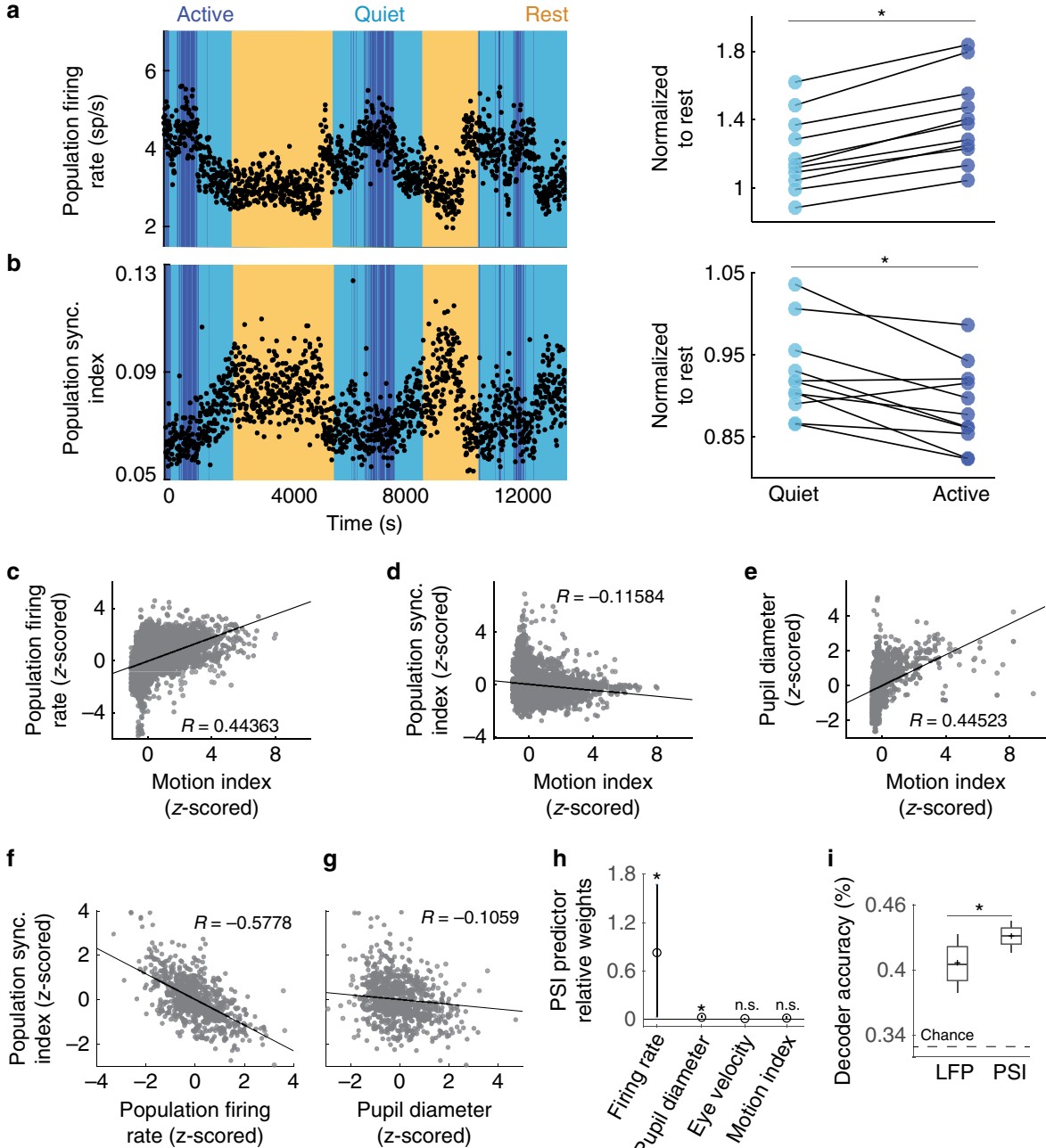

**Fig. 3 Population activity during wakefulness is influenced by behavioral state.** Population firing rate (**a**) increases and PSI (**b**) decreases in progressively activated behavioral states (rest, quiet wakefulness, and active wakefulness). An example session (left) and a summary of all sessions normalized relative to rest (right) show significant differences in cortical state between active and quiet wakefulness (asterisks indicate Friedman test $p < 0.001$ and two-sided Wilcoxon signed-rank test with Bonferroni correction $p < 0.01$, $n = 11$ sessions). Pearson's correlation demonstrates that active behavior is correlated with increased firing rate (**c**; $R = 0.44363$, $p = 1e-12$, $n = 8204$ epochs), decreased PSI (**d**; $R = -0.11584$, $p = 6.5e-26$, $n = 8204$ epochs), and increased pupil diameter (**e**; $R = 0.44523$, $p = 7.2e-106$, $n = 2164$ epochs). We performed a multiple linear regression to assess the relative contributions to PSI of population firing rate, pupil diameter, eye velocity, and motion index ($R = 0.2859$, $p < 0.001$, $n = 2164$ epochs). 2D projections of this regression show the relationships of population firing rate (**f**) and pupil diameter (**g**) to PSI. Regression coefficients are $-0.5778$ and $-0.1059$, respectively. We further performed a relative weight analysis on the predictor weights to determine which variables significantly influence PSI (**h**). Circles indicate regression coefficients, and error bars indicate a 99.9% confidence interval around the coefficients. Both population firing rate and pupil diameter are significant contributors to PSI ($p < 0.001$; 99.9% confidence interval of regression coef. (**i**) Decoder analysis reveals that PSI provides better prediction of behavioral state than LFP power ratio. Support-vector machine decoders were trained to classify ongoing behavioral state as either active wakefulness, quiet wakefulness, or rest. Decoders were trained with a random half of the data then tested with the other half, and this process was replicated 1000 times for each decoder. While both decoders perform better than chance level (33%), the PSI-trained decoder was significantly more accurate (43.1% vs. 40.6%) in predicting underlying behavioral state than the LFP-trained decoder (two-sided Wilcoxon rank-sum test, $p = 0.008$, $n = 6081$ epochs). Box indicates first and third quartiles, whiskers indicate minimum and maximum values, midline indicates median and cross indicates mean.

cortical circuits and pupil dilation[2–4,7,19–22]. To determine whether locomotion changes general arousal, animals were trained to wear a wireless eye tracker (see "Methods") which allows us to record pupil diameter and eye movements (Fig. 1b). Pupil diameter and motion index were highly correlated with behavioral state across sessions (Pearson's correlation, $R = 0.44523$, $p < 0.0001$; Fig. 3e and Supplementary Fig. 2). Furthermore, pupil diameter was positively correlated with population firing rate (Pearson's correlation, $R = 0.29972$, $p < 0.0001$) and negatively correlated with PSI (Pearson's correlation, $R = -0.27307$, $p < 0.0001$), which was apparent even in individual sessions (Supplementary Fig. 5a–d).

One possible confound that could partially explain the changes in dlPFC responses during wakefulness is eye movements (Supplementary Fig. 6; see "Methods"). Across sessions, population firing rate had a subtle yet significant positive correlation with eye velocity and saccade rate (Supplementary Fig. 7a, b; Pearson's correlation, $R = 0.16213$ and $0.12629$, respectively, $p < 0.001$). However, PSI was not significantly correlated with eye velocity ($R = -0.094515$) or saccade rate ($R = -0.092722$; Supplementary Fig. 7c, d). To determine which measure significantly influences the fluctuations in population synchrony, we performed a multiple regression relative and relative weight analysis[23], which is an important supplement to multiple regression when predictor variables are correlated among themselves, as is the case in our data. This analysis transforms predictor variables to create a new set of orthogonal predictors, which are then used for multiple regression[24]. Using this approach, we determined that PSI significantly depends on both firing rate and pupil diameter (Fig. 3f–h, $p < 0.001$; 99.9% confidence interval for the predictor weights is greater than zero), but does not significantly depend on eye velocity or motion index when all other predictors are taken into account. These results indicate that pupil diameter, population firing rate, and PSI are all features of the underlying brain state which co-fluctuate as animals transit through different behavioral states.

**Cortical states in functionally-defined cell types**. We further investigated how behavioral state impacts neuronal subpopulations by classifying well-isolated single units into narrow-spiking and broad-spiking subpopulations based on the shape of the average action potential waveform (Fig. 4a, see "Methods"). Previous studies have suggested that narrow-spiking units are likely to represent parvalbumin interneurons, whereas broad-spiking units represent a mix of mostly pyramidal cells and some interneurons[25,26]. Consistent with past work[27,28], narrow-spiking neurons comprised 16% of the overall population and had significantly higher firing rates than broad-spiking cells (Wilcoxon rank-sum test, $p < 0.01$; mean rates 5.63 sp/s and 4.02 sp/s, respectively; Fig. 4b). Firing rates of both subpopulations were influenced by behavioral state, with highest firing rates occurring in the active state and lowest in rest (Fig. 4c; $p < 0.001$, Friedman test; $p < 0.001$, Wilcoxon signed-rank test with Bonferroni correction for pairwise comparisons). Given the similar effects of behavioral state on firing rates in both subpopulations, we investigated the state dependency of the ratio between excitation and inhibition. For each epoch, we computed the ratio of average firing rates of putative excitatory and inhibitory subpopulations, referred to as "E/I ratio". Overall, there was a significant reduction in E/I ratio during rest, indicating a shift to a more inhibition-dominated regime relative to wakefulness, and a modest, but significant decrease in active relative to quiet wakefulness (Fig. 4d; $p < 0.0001$, Kruskal–Wallis; $p < 0.0001$, Wilcoxon rank-sum test with Bonferroni correction). The dynamic shift towards inhibition observed during rest implies an active role of inhibitory

subpopulations in maintaining the synchronous state. However, the differences in E/I ratio during wakefulness suggest that the responses of putative excitatory and inhibitory populations during wakefulness are complex and involved more than just cortical state dynamics.

Putative excitatory and inhibitory subpopulations were tuned to the behavioral state of the animal (see "Methods"). Population synchrony was highest during rest, and gradually decreased during quiet to active wakefulness for both subpopulations (Supplementary Fig. 8; Wilcoxon rank-sum test with Bonferroni correction; $p < 0.0001$). Surprisingly, the degree of population synchrony of the narrow-spiking population was larger than that of the broad-spiking population regardless of ongoing behavioral state (Fig. 4e–g; $p < 0.0001$, Wilcoxon rank-sum test with Bonferroni correction). Cortical inhibitory neurons are electrically connected through gap-junctions, which could underlie this enhanced synchrony[29]. Narrow-spiking units have also been reported to fire for a brief phase of the cortical slow oscillation, which could further increase low-frequency synchrony[30,31]. To determine if both subpopulations are modulated differently by quiet and active wakefulness, we normalized the PSI values in these states to their rest values. In active wakefulness, narrow-spiking neurons are substantially more desynchronized relative to rest than the broad-spiking cells ($p < 0.0001$, Kruskal–Wallis test; $p < 0.0001$, Wilcoxon signed-rank test). In the quiet wakefulness state, the broad-spiking subpopulation is slightly more desynchronized relative to rest than the narrow-spiking subpopulation (Supplementary Fig. 8; $p < 0.0001$, Wilcoxon signed-rank). Relative to their respective rest values, the PSI of broad-spiking and narrow-spiking populations showed a clear difference in their state-dependent dynamics. The broad-spiking population was highly desynchronized during both quiet and active wakefulness. The narrow-spiking cell population was only slightly desynchronized during quiet wakefulness, but strongly desynchronized during active wakefulness. These results suggest that arousal may exert a disproportionate influence on the narrow-spiking neuronal subpopulations.

## Discussion

A prominent view of sensory processing is that cortical networks are desynchronized during active wakefulness and synchronized during quiet wakefulness and sleep[1,8–12,32]. This view is based on earlier rodent studies in which technical limitations required animals to be restrained[2–4,7], hence preventing the analysis of dynamic states of cortical activity in larger animals freely moving in their environment. Furthermore, whether the impact of behavioral state on cortical population activity extends to higher-order, executive cortical areas has been unknown. We overcame technical limitations inherent in previous studies by developing a system that integrates wireless transmission of cortical activity, wireless eye-tracking, and real-time video analysis to examine the dynamics of population activity in dlPFC of unrestrained monkeys. Our approach allowed us to identify distinct cortical sub-states during quiet and active wakefulness, and transitions in population activity during drowsiness and rest. These results reveal that wakefulness is not a purely desynchronized state, rather synchrony during wakefulness co-fluctuates with behavioral demands.

Previous studies performed in rodent sensory cortex have established a relationship between the degree of ongoing cortical synchrony and behavioral performance[13]. However, whether locomotion in unconstrained nonhuman primates yields reliable shifts in cortical state, as observed in rodents, has been unknown. Furthermore, previous studies of cortical state have focused on sensory areas and stimulus coding[2–4,7], but whether the results

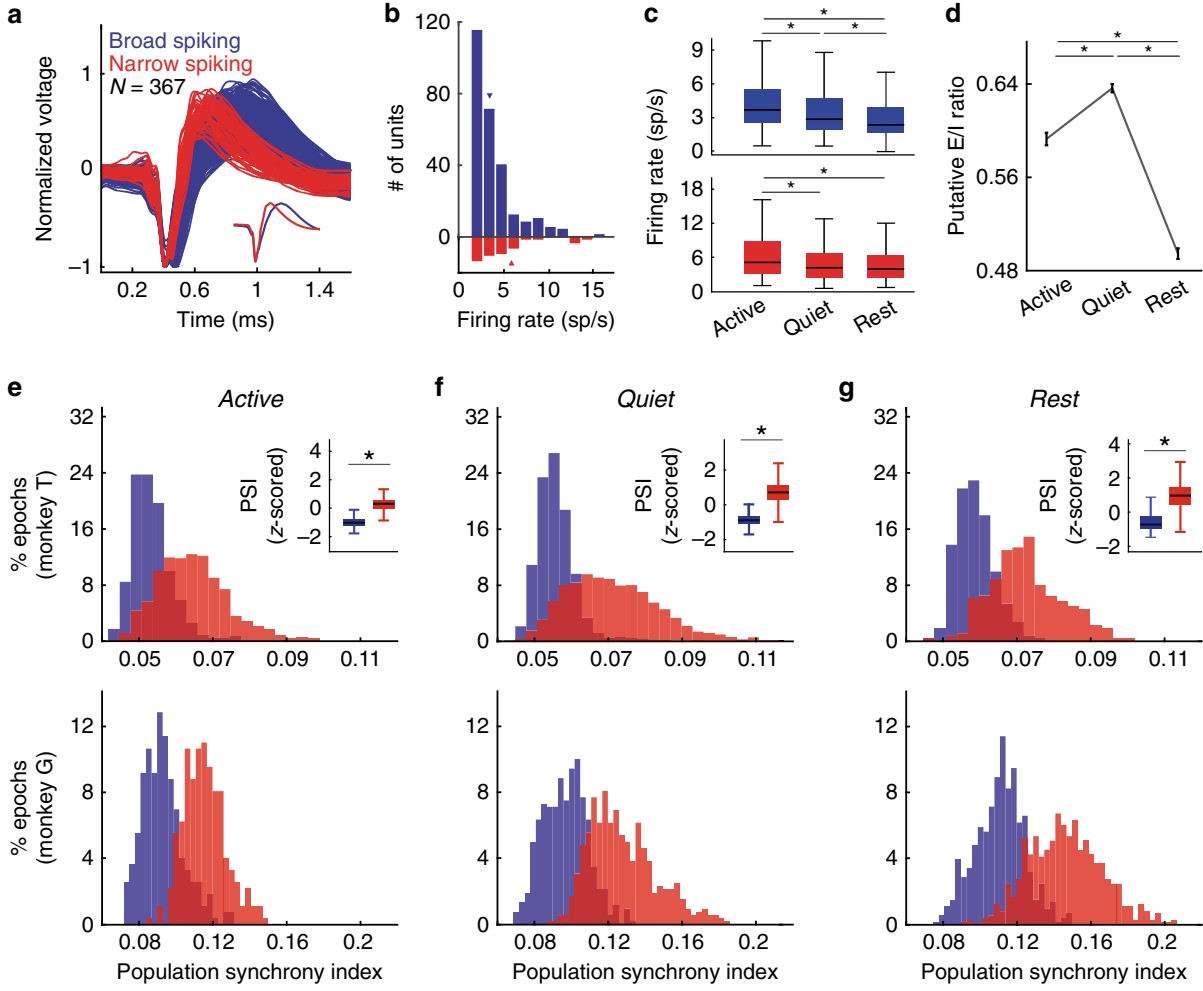

**Fig. 4 Putative inhibitory populations are more synchronous than putative excitatory populations. a** Mean spike waveforms of 367 recorded single units classified as narrow-spiking or broad-spiking based on peak-to-trough duration. **b** Narrow-spiking units had a significantly higher mean firing rate than broad-spiking units (5.63 sp/s vs. 4.02 sp/s; $p = 0.0023$, two-sided Wilcoxon rank-sum, $n = 367$ U). **c** (Top) Mean firing rates of broad-spiking units during active, quiet, and rest states. Firing rates are highest in the active state, and lowest in the rest state, and at an intermediate level in the quiet state ($p = 4.1e{-}5$; Friedman test; asterisks indicate $p < 0.0001$; two-sided Wilcoxon signed-rank test with Bonferroni correction; $n = 308$ broad-spiking units). (Bottom) Same analysis for narrow-spiking units. Firing rates are significantly higher in the active state than either the quiet or rest state ($p = 1.5e{-}4$; Friedman test; asterisks indicate $p < 0.0001$; two-sided Wilcoxon signed-rank with Bonferroni correction; $n = 59$ narrow-spiking units; the reduction in firing rates between the quiet and active states is not significant, $p = 0.01$). Boxplot box indicates first and third quartile, center line of the box indicates the median, and whisker lengths reflect the inter-quartile range multiplied by 1.5. **d** Putative E/I ratio computed as the ratio of broad-spiking to narrow-spiking firing rates across all behavioral states. E/I ratio slightly increases in quiet relative to active state, however there is a particularly strong swing towards inhibition during rest ($p < 0.0001$, Kruskal–Wallis test; asterisks indicate $p < 0.0001$, two-sided Wilcoxon rank-sum test with Bonferroni correction; error bars indicate mean ± standard error, $n = 8204$ epochs). **e–g** PSI was computed for both narrow-spiking and broad-spiking populations in active (**e**), quiet (**f**), and rest (**g**) states. Broad-spiking populations for each session were subsampled 100 times to match the population size of the narrow-spiking units and PSI was averaged overall sub-samplings. Broad-spiking PSI and narrow-spiking PSI distributions are shown for both monkey T (top) and monkey G (bottom). PSI for both subpopulations was z-scored and combined across both monkeys for each behavioral state, demonstrating significantly higher PSI for the narrow-spiking population in all behavioral states (top, insets; asterisks indicate $p < 0.0001$; two-sided Wilcoxon signed-rank test).

can be extended to executive areas, such as dlPFC, has been unclear. Our results reveal that the impact of brain state on cortical dynamics, including population synchrony and firing rate, are remarkably similar across brain areas and model systems. This indicates that the main features of cortical state have remained evolutionarily unchanged across species, which further supports the idea that they must convey substantial functional advantages to the organism.

Our study rests on our successful development and integration of three key technologies—wireless eye tracking, wireless multi-electrode recording, and behavioral monitoring—in a focused effort to advance our understanding of the dynamics of cortical

state in freely-moving nonhuman primates. However, our approach has a number of technical limitations that will need to be addressed in future work in order to completely characterize cortical states and their relationship to behavior. First, our wireless eye tracker cannot resolve microsaccadic activity inherent during visual fixation, and hence we cannot establish links between fixational eye movements and ongoing cortical state. Second, the fact that our electrophysiological recordings are limited to one cortical area, dlPFC, makes it impossible to investigate how changes in brain state impact the communication between brain areas. Third, electrophysiological classification of functional cell types based on spike waveforms does not capture

the heterogeneity of genetically-defined neuronal subtypes. Our electrophysiological classification of waveforms may result in a narrow-spiking population of likely parvalbumin-expressing interneurons, and a broad-spiking population mostly comprised of excitatory pyramidal cells and a subset of other interneurons subtypes. Finally, our chronic electrode arrays do not allow us to record across all the layers of the cortical column, and hence we cannot investigate how cortical state operates within the depth of cortex.

We have demonstrated that arousal is a key determinant of population activity in dlPFC, strongly linked with locomotor behavior. Furthermore, regardless of behavioral state, narrow-spiking cells exhibit stronger synchronized fluctuations in population activity compared to broad-spiking neurons. These results imply that arousal influences network behavior mainly by modulating the synchronized fluctuations of putative parvalbumin inhibitory cells[32]. The greater degree of population synchrony in these cells may serve to limit low-frequency synchrony in excitatory cell populations and maintain stable firing rates across behavioral states[33]. Our integration of multiple wireless recording methodologies has allowed us to study cortical function during complex behavioral states that are difficult or impossible to observe in an experimental rig. This approach opens up new opportunities for future large-scale wireless electrical recordings in many brain areas coupled with advanced behavioral monitoring to explore questions that have been unapproachable until now.

## Methods

**Ethics statement**. All experiments were performed in accordance with protocols approved by the Animal Welfare Committee and the Institutional Animal Care and Use Committee for the University of Texas Health Science Center at Houston (UTHealth).

**Surgical procedures**. A titanium headpost was implanted medially with anchor screws. Following a recovery period exceeding 6 weeks, animals were acclimatized to the experimental cage for at least 4 days per week for over 4 weeks. After acclimatization, animals were implanted with a 96-channel Utah array in the left dlPFC (area 46) and pedestal on the caudal skull (Blackrock Microsystems). The stereotaxic coordinates were chosen to make the craniotomies based on MRI and brain atlases[15,34,35]. During surgery, visual identification of arcuate and principal sulci guided precise implantation of arrays into the dlPFC. Following array implantation, animals had a 2-week recovery period before recording from the array.

**Behavioral paradigm**. Two 9-year-old adult male rhesus monkeys (*Macaca mulatta*) were used for these experiments. Recordings were performed in a custom $4' \times 2' \times 3'$ (LxWxH) plastic cage surrounded by an array of eight antennae. To encourage walking around the cage, animals were trained to push two buttons on either side of the cage to receive a small food reward. After ~1 h, animals were encouraged to rest by suspending food access and turning down the lights. IR illumination was used to monitor the animals when lights were off. Only sessions in which the animals took a rest were included in analysis. Recording sessions were 2 h and 50 min on average, and we were able to record for up to 6 h. Animals rested for 44 min per session on average. During a subset of sessions in which the eye tracker was worn, animals were not given a lights-out period. The animal's natural rest postures would interfere with the eye tracker, potentially causing damage to the device. Furthermore, the eye tracker was not deemed necessary for rest analysis as eyelids are closed during rest.

**Wireless electrophysiology**. Following a postsurgical recovery period, we wirelessly recorded from the 96-channel multielectrode array in dlPFC (see surgical procedures). Cereplex W (Blackrock Microsystems) was attached to the pedestal and the animal was transferred into the plastic recording cage, which is $4' \times 2' \times 3'$ (LWH), and surrounded by eight directional antennae. Physiological data was recorded on a Cerebus neural signal processor (Blackrock Microsystems). Continuous LFP was recorded at 2 kHz and was used for spectral analysis. Analysis of LFP spectra were implemented using an open-source MATLAB package (Chronux). Putative spike waveforms were captured upon threshold crossing at 30 kHz and sorted offline using Plexon Offline sorter. Analyses were done using only well-isolated single units. Each well-isolated single unit was either classified as a putative narrow-spiking interneuron or broad-spiking pyramidal cell. For each unit, spike waveforms were averaged, normalized, and spline interpolated to a 2.5 μs

resolution[27]. We then computed the time between peak and trough, and used a threshold of 300 μs to classify units as putative interneurons and pyramidal cells[28].

**Behavior tracking**. We recorded a top-down video of the animals during the experiments using an IR-sensitive camera. Video frame acquisition was controlled by a dedicated computer running custom Python scripts to send a TTL pulse to be recorded by the DSP upon acquisition of each frame in order to synchronize video frames with neural and eye tracker data. During the experiment, access to food would temporarily be suspended and lights turned off to encourage resting. Epochs of rest were defined as any period during the experiment in which the animal entered and maintained a resting posture for a period of at least 5 min. The video for the wake periods was concatenated and processed to quantify the animals' movement. The motion index was defined as the average number of pixels that change in intensity between consecutive frames. The recordings were binned into 10 s, nonoverlapping epochs. Wake epochs were classified as either active or quiet based on whether the motion index was greater than or less than the average motion level during wakefulness. Motion index was computed using custom software written in Python using the OpenCV package.

**Eye tracking**. We used a custom wireless eye tracker (ISCAN) to measure pupil position and diameter during sessions in which animals were not encouraged to rest. Pupil diameter was recorded using the neural signal processor (Blackrock microsystems) and synchronized to the overhead video in the same way as the neural data (see Behavioral tracking). Pupil diameter and position were sampled at 1 kHz. For the sessions in which pupil diameter was recorded, the animals were not encouraged to rest to prevent any damage or movement of the eye mirror. To train animals to wear the device without damaging it, its 3D geometry was modeled (Sketchup Pro), and dummies were 3D printed and fitted with eye mirrors. To properly position the eye tracker and dummies relative to the eye, custom adapters were designed, and 3D printed to attach directly to the headpost and serve as an anchor point for the eye tracker. These adapters were designed to interface with the headpost, without touching the animal directly, to minimize discomfort, and reduce the likelihood of the device being tampered with. These dummy eye trackers were worn by animals for several mock recording sessions to adjust them to wearing the device. Once the animals grew accustomed to wearing the dummy and stopped touching it altogether, the real device was used. Horizontal and vertical coordinates of the pupil were recorded and used to compute eye velocity. To extract fixations, eye velocity was thresholded at one standard deviation above the median. Any period of time greater than 5 ms during which the eye was below this threshold was considered a fixation.

**Population synchrony index**. Spike rasters were separated into 10 s epochs and binned into 10 ms windows. For each 10 ms window, the average firing rate for the population was computed. The relationship between population firing rate and behavioral state was determined by averaging population firing rate within each 10 s epoch to match the resolution at which behavioral state was defined (see Behavioral tracking). The PSI for each 10 s epoch was computed from the corresponding 10 ms resolution population average firing rate trace. This trace was Fourier transformed for each epoch to yield a set of Fourier coefficients. PSI for a given epoch was given by the average of Fourier coefficients for frequencies between 0.5 and 10 Hz, divided by the mean population firing rate for the epoch. PSI quantifies the magnitude of low-frequency oscillations in the population spiking activity. For the analyses involving putative broad-spiking and narrow-spiking populations, PSI was computed separately. Sessions with fewer than four narrow-spiking subpopulations were excluded from these analyses. PSI for narrow-spiking subpopulations was computed as described above. For the broad-spiking population, bootstrapping was employed to control for any influence of population size on PSI values. From the overall broad-spiking population, 100 subpopulations were selected randomly for each session such that each broad-spiking subpopulation size was equal to the size of the narrow-spiking population. PSI was computed for all epochs for each broad-spiking subpopulation. PSI of the broad-spiking population in a given epoch was given by the average PSI value overall 100 subpopulations. For analyses pertaining to Supplementary Fig. 4b, a set of populations of units within a radial distance were determined based on the location of the electrode detecting each unit within the geometry of the multielectrode array. Populations with fewer than 4 U were not used in this analysis. Comparisons of PSI across different states were made using the same populations of units across each state.

**Statistics and reproducibility**. Statistical significance was assessed using nonparametric tests. Wilcoxon signed-rank and Wilcoxon rank-sum tests were used when comparing two groups of unmatched or matched observations, respectively. When significant differences among three groups (active, quiet, and rest) were assessed, Kruskal–Wallis and Friedman tests were used to detect differences in unmatched and matched data, respectively. In these cases, post-hoc pairwise tests were conducted if differences among groups were determined. Because we did not want animals to sleep while wearing the eye tracker, we analyzed 11 sessions that included a rest period and did not include eye tracking and we analyzed a further nine sessions that included the eye tracker and no rest periods. We additionally

employed a support-vector machine decoder to determine how well PSI and LFP could predict behavioral state. Decoders were trained with data combined across all sessions. Each session contributed the equal epochs of each behavioral state, and this number was equal to the number of occurrences of the least frequent state for each session. LFP power ratio and PSI were z-scored within each session before combining data across all sessions. We employed tenfold cross validation to ensure that the SVM was appropriate (i.e., not overfitting to the specific training data), and then continued by training the decoder with a random half of the data set and testing with the other half. This train-half, test-half paradigm was repeated 1000 times, and average performance was reported. We ran a multiple regression with relative weight analysis to determine how different factors impact population synchrony. This relative weight analysis was performed in R using an online tool (RWA-Web) to account for correlations between predictor variables of multiple regression analysis[24]. All other statistical analyses were performed in MATLAB.

**Reporting summary**. Further information on research design is available in the Nature Research Reporting Summary linked to this article.

## Data availability
The data used in this study are available from the corresponding author upon reasonable request.

## Code availability
The code used for data analyses in this study is available from the corresponding author upon reasonable request.

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

## Acknowledgements
This work was supported by awards from the NIH BRAIN Initiative.

## Author contributions
R.M. and V.D. designed the experiments. R.M. and N.S. performed the experiments. R.M. analyzed the data. R.M. and V.D. wrote the paper.

## Competing interests
The authors declare no competing interests.
