## [Peer Review File · Nature Communications]

Reviewers' Comments:

Reviewer #1:

Remarks to the Author:

This study explores an interesting issue in neuroscience, whether they are “states” of cortical activity in executive areas of the prefrontal cortex of non-human primates during wakefulness and sleep. The authors used a state-of-the-art recording system in freely moving macaque monkeys to record the activity of neurons in the lateral prefrontal cortex. The novel findings in this study are the cortical area where recordings were conducted, the lateral prefrontal cortex, and the species, the macaque monkey. One additional finding is that narrow spiking neurons, likely putative parvalbumin interneurons seem to synchronize their responses to a larger degree than broad spiking neurons. This is an interesting paper that makes a novel contribution to the field. As I said earlier, the technology development is very novel. I am amazed the eye tracker works and the animal does not remove it, it seems that at some point would interfere with the visual field and these animals have the habit to explore the implants. How did the authors deal with this problem? Did they train the animals? Is this a technique that requires some additional training or only certain animals can participate in the study?

I have some comments that if addressed could improve the quality of the paper. See below:

1. It is unclear whether the rest period has no lights and the active period had lights. If so, can the results be explained by the differences in lighting conditions? Do the authors have some controls that can show this is not the case?
2. Can the authors provide a rationale about how were the frequency bands defined to compute the ratio?
3. Figure 1d is not clear how population firing rate was computed
4. Can the authors show some eye position traces with saccades? One potential issue here is that during natural stimulation the eye movements change their distribution relative to head fixed animals. Do the authors see different types of eye movements and do they have the resolution to measure them? (e.g., OKN, smooth pursuit, saccades, VOR) Did they also measure head movements?
5. Locomotion and pupil diameter seem correlated. It may be plausible that it is arousal rather than self motion what modulates neuronal activity. Here it seems to me appropriate to use multiple regression techniques or GLM to tease apart the contribution of the different factors to modulate neural activity. The authors did conduct some of this analysis but for the eye movements and pupil diameter.
6. The suggestion that narrow spiking neurons are likely inhibitory interneurons seems reasonable, but the suggestion that broad spiking neurons are putative excitatory neurons is not fully justified. This seems to be a misconception in the field (see Casale et al. 2015 for an example). I would suggest the authors modify the language in this part of the manuscript. That said, it is very interesting that the narrow spiking neurons, likely Parvalbumin interneurons show a stronger degree of synchrony/modulation relative to broad spiking ones.
7. The discussion is very short, it needs to be expanded. The authors should explain the importance of their results, not only from the technical view point but to our knowledge of how primate brain networks work. For example, is there any difference between the way sensory cortices encode behavioral states and the way the PFC does?
8. The authors need to provide a better location of the arrays, I suggest using the anatomical maps of the PFC by Petrides (2005) and refer to the area as 8A or 9/46. This needs to be presented in more detail. It is important for the purpose of replicability. The transition between behavioral states is also a very interesting part of this work, is it a gradual transition or is it abrupt?
9. Can the authors use encoding or decoding models to derive behavioral state from the set of variables used here? For example, can neuronal activity predict behavioral state? They seem to have a

substantial amount of data, so using decoders may be a reasonable way to do it.

Reviewer #2:

Remarks to the Author:

This manuscript examines the dynamics of neural activity in the dorsolateral prefrontal cortex of unrestrained, normally active monkeys. It is commonly accepted that cortical networks switch from a state in which they exhibit synchronized responses during sleep to a state of desynchronized activity during active wakefulness. Evidence at the level of single cell resolution has come primarily from studies that recorded from sensory areas in restrained rodents. However, little is known about whether these changes in dynamics extend to higher-order executive areas of primates.

In an impressive tour de force, Dragoi and coworkers have succeeded in integrating methods for recording neural activity, measuring eye movements, and monitoring motor behavior, as monkeys move normally within a confined environment. They show that changes in synchronization extend to executive cortical areas and that the degree of desynchronization during wakefulness depends on the amount of motor activity. Furthermore, they show that irrespective of the behavioral state, the degree of synchronization appears to be stronger in inhibitory neurons than in excitatory ones, suggesting a possible mechanism by which arousal may operate.

I am impressed by this work and think it can be published essentially in its current version. I find the results clear and convincing and believe their publication will set a new standard that other researchers will try to achieve. I do not have comments other than a few clarification points below.

It may be worth expanding a bit the discussion, as one may wonder whether the observed changes in cortical states may be caused by behavioral factors beyond the level of resolution provided by the experimental setup. For example, one question would be if the small incessant eye movements made by monkeys, which are difficult to detect with the head-mounted device used in this study, could have exerted an influence. However, these are questions to be addressed in future studies. Given the novelty and degree of technical innovation of this work, they can here be mentioned in the discussion.

There are also a few points in which clarity could be improved. The main point regards the panels in Fig. 3c-g. What is the rationale for showing all these correlations? Although significant, some of these correlations appear weak and do not account well for variance. I guess I am missing the main messages that the reader should take from this section.

Also, is there a real need for using Friedman tests in Figure 3a and b?

Reviewer #3:

Remarks to the Author:

I find the manuscript entitled "Dynamic states of population activity in prefrontal cortical networks of freely-moving macaque" by Milton et al., certainly of interest to the broad audience of Nature.

The paper is mostly well written and appropriately documented. The findings are somewhat expected yet novel. "...that wakefulness is not a purely desynchronized state, rather synchrony during wakefulness co-fluctuates with behavioral demands..." and the implied modulations by the synchronized fluctuations of inhibitory cells..." was suggested by Steriade and others but is here well documented and experimentally shown in awake NHPs. I find this to be a big strength for the paper,

as is the necessary development and implementation of an effective wireless recording system concomitant with eye tracking, behavioral assessment, etc. So, in short, I recognize this as a tour de force in this species, and, this, to some extent, compensates the problem of the very low n (n=2).

I find the paper publishable with minor changes, of which I strongly suggest the following:

Please revise accordingly so that the Results start with results and not with narrative of the methods. I understand space limitations for this type of communication yet, it is very important that the authors discuss the limitations of the study. Among these, the authors should comment on the low "n" and the fact that both animals were adult males; if possible the age should be made available. Would it be possible that the observations and measurements changed as a function of gender and/or age? The authors should clearly state the number of replicas, recording sessions, number of epochs of the same activity that were lump together for analysis, etc. The statistical methods appear adequate yet the choices are not really explained to any extent.

Very respectfully,
Alvaro Duque, Ph.D.

Reviewer #1

Specific comments:

1. It is unclear whether the rest period has no lights and the active period had lights. If so, can the results be explained by the differences in lighting conditions? Do the authors have some controls that can show this is not the case?

We believe the reviewer is referring to the quiet wakefulness period, not rest (when eyes are closed and hence lighting conditions are irrelevant). To clarify, lights were turned off prior to rest in order to encourage animals to take a nap, and the epochs of quiet wakefulness during the lights-off period right before rest were not included in the analyses. Therefore, there were no differences in lighting conditions between active and quiet wakefulness. Occasionally, animals decided to nap while the lights were still on. However, these resting periods were indistinguishable from the lights-off standard condition (no differences in mean population synchrony and firing rates. The text of the Results section has been updated to improve clarity. In paragraph 1 of the Results, the explanation of behavioral state classification has been updated to more clearly indicate the lighting conditions throughout the recording sessions. Additionally, we clarify that lighting conditions were invariant during the analyzed periods of wakefulness: “*Periods when the animal was awake, but the lights were off were not included in subsequent analyses*” (page 4, paragraph 1).

2. Can the authors provide a rationale about how were the frequency bands defined to compute the ratio?

Previous literature examining global states in rodent models have established that cortical activation during wakefulness and arousal are linked to a significant reduction in low-frequency LFP power (<10 Hz) and an increase in high-frequency LFP power (>20 Hz) (e.g., Niell and Stryker, *Neuron*. 2010; Li et al. *Science*. 2009; Poulet and Crochet, *Front. Syst. Neurosci.* 2019). We computed the LFP power ratio using these frequency bands because they have previously been demonstrated to reflect changes in the global brain state. We have made changes to the 2nd paragraph of the Results section to better explain the rationale behind the frequency bands used.

3. Figure 1d is not clear how population firing rate was computed

Population firing rate was computed in each 10 s epoch by taking the total number of spikes recorded across the entire population within each 10 ms bin, then converting to spikes per unit dividing by the number of units and converting to spikes/second by multiplying with a scaling factor of 100. We now clarify how population firing rate was computed in the 2nd paragraph of the Results section, as well as in the Methods section and in the figure caption for **Figure 1d**.

4. Can the authors show some eye position traces with saccades? One potential issue here is that during natural stimulation the eye movements change their distribution relative to head fixed animals. Do the authors see different types of eye movements and do they have the resolution to measure them? (e.g., OKN, smooth pursuit, saccades, VOR) Did they also measure head movements?

An example eye position trace with saccades has been included as a new supplemental figure (**Supplementary Fig. 6**). The distributions of different types of eye movements would certainly be expected to differ between head-fixed and freely-moving animals. However, examining (and comparing) the impact of head restraint was beyond the scope of this work. Our system for recording the eye and tracking animal behavior is optimized for distinguishing between saccades and fixations by thresholding

eye velocity. Future higher resolution versions of our eye tracking and behavioral monitoring system will be better suited towards identifying all types of eye movements in order to investigate how they change in different behavioral states. Furthermore, future eye tracking systems will be optimized to precisely measure head position in 3D space, which is important for distinguishing between smooth pursuit and VOR movements. Current limitations and future potential of the wireless eye tracker for examining different types of eye movements is now briefly discussed in the 3rd paragraph of the Discussion section.

5. Locomotion and pupil diameter seem correlated. It may be plausible that it is arousal rather than self motion what modulates neuronal activity. Here it seems to me appropriate to use multiple regression techniques or GLM to tease apart the contribution of the different factors to modulate neural activity. The authors did conduct some of this analysis but for the eye movements and pupil diameter.

The multiple regression analysis previously shown in **Figure 3h** regressed PSI from population firing rate, pupil diameter, and eye velocity. This analysis has been updated to include motion index as an additional predictor variable. We found that when population firing rate, pupil diameter, eye velocity, and motion index are all considered, only firing rate and pupil diameter significantly improve the regression of PSI. This result supports the idea that the underlying mechanism by which locomotion impacts cortical state involves the global arousal circuitry because motion index does not improve the regression when pupil diameter is also used as a predictor variable. This is now discussed on page 7, 3rd paragraph.

6. The suggestion that narrow spiking neurons are likely inhibitory interneurons seems reasonable, but the suggestion that broad spiking neurons are putative excitatory neurons is not fully justified. This seems to be a misconception in the field (see Casale et al. 2015 for an example). I would suggest the authors modify the language in this part of the manuscript. That said, it is very interesting that the narrow spiking neurons, likely Parvalbumin interneurons show a stronger degree of synchrony/modulation relative to broad spiking ones.

The language has been modified as suggested throughout the manuscript, and the limitation of the waveform-based classification procedure has been included in the 3rd paragraph of the Discussion section. For instance, one relevant portion states that: “*electrophysiological classification of cell types based on spike waveforms does not capture the heterogeneity of genetically-defined neuronal subtypes. Our electrophysiological classification of waveforms results in a narrow-spiking population of likely parvalbumin-expressing interneurons, and a broad-spiking population that is comprised mostly of excitatory pyramidal cells with a subset of other interneurons subtypes*” (page 10, 2nd paragraph).

7. The discussion is very short, it needs to be expanded. The authors should explain the importance of their results, not only from the technical view point but to our knowledge of how primate brain networks work. For example, is there any difference between the way sensory cortices encode behavioral states and the way the PFC does?

We have expanded the Discussion section to highlight the importance of our results both from the technical viewpoint but also in the context of previous work in sensory cortex. Additionally, we now discuss the limitations of our technique (pages 9-11).

8. The authors need to provide a better location of the arrays, I suggest using the anatomical maps of the PFC by Petrides (2005) and refer to the area as 8A or 9/46. This needs to be presented in more detail. It is important for the purpose of replicability. The transition between behavioral states is also a very interesting part of this work, is it a gradual transition or is it abrupt?

We have updated the introduction and methods sections to include the stereotactic location (area 46) along a reference to the anatomical maps we used (Paxinos, 2000). Additionally, we have added a supplemental figure (**Supplementary Fig. 3**) that shows that behavioral and cortical state transitions tend to closely track each other when the animal transitions between active and quiet wakefulness. A more in-depth analysis of transitions is discussed as a potential pathway for future work.

9. Can the authors use encoding or decoding models to derive behavioral state from the set of variables used here? For example, can neuronal activity predict behavioral state? They seem to have a substantial amount of data, so using decoders may be a reasonable way to do it.

This is a great suggestion – we followed reviewer’s advice and decoded behavioral state from neuronal activity. Results have been included in a new panel (**Fig. 3i**). Specifically, we trained support-vector machines to use LFP power ratio or PSI to predict whether the behavioral state was active wakefulness, quiet wakefulness, or rest. We found that, while both metrics can predict behavioral state better than chance level, the decoder performs better when using PSI to distinguish between behavioral states. The 5th paragraph of the Results section has been modified to reflect this new result (page 6, top).

Clarification points:

1) I am amazed the eye tracker works and the animal does not remove it, it seems that at some point would interfere with the visual field and these animals have the habit to explore the implants. How did the authors deal with this problem? Did they train the animals? Is this a technique that requires some additional training or only certain animals can participate in the study?

The reviewer is correct in assuming that animals would explore the eye tracker, which could cause damage to the device. We prevented this by fitting animals with dummy eye trackers which animals were free to explore until they lost interest. After animals were accustomed to wearing the dummy and no longer investigated it, the dummy was swapped with the real eye tracker without incident. An explanation of the training procedure has been included in the 1st paragraph of the Results section: “Animals were initially fitted with a dummy device until they learned to ignore the lens in their field of view and lost interest in touching and potentially damaging the head-mounted eye tracker. Once the animals could consistently wear the dummy eye tracker without touching it, the dummy was swapped with the real device which the monkeys wore without damaging”.

Reviewer #2

1) It may be worth expanding a bit the discussion, as one may wonder whether the observed changes in cortical states may be caused by behavioral factors beyond the level of resolution provided by the experimental setup. For example, one question would be if the small incessant eye movements made by monkeys, which are difficult to detect with the head-mounted device used in this study, could have exerted an influence. However, these are questions to be addressed in future studies. Given the novelty and degree of technical innovation of this work, they can here be mentioned in the discussion.”

This is an important issue – the 3rd paragraph of the Discussion section has been added to focus on questions for future studies and limitations of our current work. As the reviewer points out, our current eye tracker is unable to detect microsaccades during fixation. We discuss this, and several other possibilities for future investigations, in the 3rd paragraph of the Discussion section.

2)” *There are also a few points in which clarity could be improved. The main point regards the panels in Fig. 3c-g. What is the rationale for showing all these correlations? Although significant, some of these correlations appear weak and do not account well for variance. I guess I am missing the main messages that the reader should take from this section.*”

We have updated the text to better clarify the rationale behind these panels. In **Figure 3c-e**, we compute correlations between motion index and population firing rate, PSI, and pupil diameter. The rationale was to demonstrate that the differences between active and quiet wakefulness are not an artifact of the specific motion index threshold used to define these distinct states. Paragraph 7 of the Results section was altered to clarify this issue. In **Figure 3f-g**, we computed a multiple regression to examine how well population firing rate, pupil diameter, and eye velocity together predict PSI. Panels f and g are projections of this higher-dimensional regression analysis. The relative weights of these predictors reflect how much additional information about PSI can be gained from including the predictor when all the other variables are accounted for. The multivariable regression used in **Figure 3f-h** has been modified to include motion index as a predictor variable, as was suggested by another reviewer (see page 7, paragraph 3).

3) *“Also, is there a real need for using Friedman tests in Figure 3a and b?”*

The analysis in **Figure 3a-b** uses the Friedman test to determine if there are differences in the underlying distributions of the population firing rate and PSI across three matched groups. This test is the non-parametric analog of the repeated-measures ANOVA, however it is relatively advantageous because non-parametric tests do not assume that the data come from a normal distribution. Pairwise differences between quiet and active wakefulness were determined by a post-hoc Wilcoxon signed-rank test with a Bonferroni correction for multiple comparisons. This post-hoc test is represented in the right panels of **Figure 3a-b**. A more thorough explanation of statistical methods used has been included in the Methods section and in the 2nd paragraph of the Results section (page 5).

Reviewer #3

Minor changes:

1) *Please revise accordingly so that the Results start with results and not with narrative of the methods.*

We appreciate reviewer’s suggestion. However, in our opinion, the development of an integrated wireless system for examining brain states in freely-moving monkey represents a significant result in itself regardless of the actual electrophysiological experiments. Therefore, we were inclined to keep the description of methods that were developed as part of this investigation in the Results section. To improve clarity, we have restructured the Results section and included subheadings. The description of methods developed specifically for this work are described in the first subheading of the Results section: Integrated system to examine brain states in freely-moving macaques. Having said that, if the reviewers and editor feel strongly that the Results section should strictly start with the presentation of our electrophysiological results rather than the presentation of our technology and its implementation, we would be happy to restructure this section in a future revised manuscript.

2) *The authors discuss the limitations of the study. Among these, the authors should comment on the low “n” and the fact that both animals were adult males; if possible the age should be made available.*

Would it be possible that the observations and measurements changed as a function of gender and/or age?

The Discussion section now mentions several limitations of our present approach (page 10, paragraph 2). The age of both monkeys is now provided in the Methods section (page 22, paragraph 3). The reviewer further brought up concerns about the “low n” and the fact that both animals were male. However, studies using electrophysiological recordings from awake nonhuman primates typically use 2 animals due to the complexity involved in the recordings and the multiple sessions for each animal. Furthermore, we are unaware of any studies reporting gender differences in basic neurophysiological phenomena in sensory and executive cortical areas.

3) The authors should clearly state the number of replicas, recording sessions, number of epochs of the same activity that were lump together for analysis, etc.

A total of 11 sessions were recorded in two animals. In total, we analyzed 2027 epochs of active wakefulness, 6182 epochs of quiet wakefulness, and 2943 epochs of rest. This information has been added to the 1st paragraph of the Results section.

4) The statistical methods appear adequate yet the choices are not really explained to any extent.

We apologize for the lack of additional information on our choice of statistical methods. The rationale for our methodological choice has been added in the Results and Methods sections. A ‘Statistical analysis’ subsection of the Methods has been added, and the following explanation was added to the second paragraph of the Results section: “*Significance of results were determined with non-parametric statistical tests which do not assume an underlying normal distribution. Specifically, we used Wilcoxon signed-rank and Wilcoxon rank-sum for comparison of two groups of unmatched and matched observations, respectively. When significant differences among three groups (active, quiet, and rest) were assessed, Kruskal-Wallis and Friedman tests were used to detect differences in unmatched and matched data, respectively.*” (page 5, paragraph 1)

Reviewers' Comments:

Reviewer #1:

Remarks to the Author:

The authors have addressed my main concerns. The manuscript has improved relative to the previous version. I have no further concerns.

Reviewer #2:

Remarks to the Author:

The authors have satisfactorily addressed my previous concerns. I do not have further comments. I think this is a very interesting study that will receive considerable attention.

Reviewer #3:

Remarks to the Author:

I have read the responses to my queries and found them to be adequate. I also read the new version of the paper and find it acceptable for publication as is. I do not have any further comments.

Alvaro Duque, PhD.